# Synergistic Effects of Scalp Acupuncture and Repetitive Transcranial Magnetic Stimulation on Cerebral Infarction: A Randomized Controlled Pilot Trial

**DOI:** 10.3390/brainsci10020087

**Published:** 2020-02-07

**Authors:** Jae-Hong Kim, Jae-Young Han, Min-Keun Song, Gwang-Cheon Park, Jeong-Soon Lee

**Affiliations:** 1Department of Acupuncture and Moxibustion Medicine, College of Korean Medicine, DongShin University, Naju City 58245, Korea; nahonga@hanmail.net; 2Clinical Research Center, DongShin University Gwangju Korean Medicine Hospital, 141, Wolsan-ro, Nam-gu, Gwangju City 61619, Korea; smailcc@nate.com; 3Department of Physical and Rehabilitation Medicine, Chonnam National University Medical School and Hospital, Gwangju City 61469, Korea; drsongmk@daum.net; 4Department of Nursing, Christian College of Nursing, Gwangju City 61662, Korea; mishilee@ccn.ac.kr

**Keywords:** cerebral infarction, acupuncture, transcranial magnetic stimulation, randomized controlled trial

## Abstract

This study investigated the synergistic effects of scalp acupuncture (SA) and repetitive transcranial magnetic stimulation (rTMS), known to be effective for cerebral infarction. This outcome-assessor-blinded, randomized controlled clinical trial included a per-protocol analysis to compare the efficacy of SA and electromagnetic convergence stimulation (SAEM-CS) and single or no stimulation. The trial was conducted with 42 cerebral infarction patients (control group, 12; SA group, 11; rTMS group, 8; SAEM-CS group, 11). All patient groups underwent two sessions of CSRT per day. SA, rTMS, and SAEM-CS were conducted once per day, 5 days per week, for 3 weeks. The primary outcome was evaluated using the Fugl–Mayer assessment (FMA). FMA Upper Extremity, FMA total, MBI, and FIM scores significantly increased in the rTMS group compared with the control group. Additionally, FMA Upper Extremity, FMA total, MBI and FIM scores significantly increased in the rTMS group compared with the SAEM-CS group. However, there were no significant changes in the SA or SAEM-CS groups. In conclusion, low-frequency rTMS in the contralesional hemisphere may have long-term therapeutic effects on upper extremity motor function recovery and improvements in activities of daily living. SAEM-CS did not show positive synergistic effects of SA and rTMS.

## 1. Introduction

Stroke is the second most common cause of death and the leading cause of adult disability worldwide [1]. Cerebral infarction (CI) is a common disease with high mortality, recurrence, and disability rates, which accounts for approximately 70% of strokes [2]. Conventional treatment of stroke patients includes pharmacological treatments, surgery, and multiprofessional rehabilitation. These treatments can promote recovery to some extent; however, no single intervention clearly and definitively contributes to stroke recovery. Therefore, stroke treatment strategies should combine multiple disciplines such as neurology, rehabilitation medicine, and traditional medicine [3,4].

Neural plasticity refers to the ability of the brain to develop new neuronal connections, acquire new functions, and compensate for impairments. These processes are crucial for motor recovery after stroke [5,6,7]. Current research aims to determine whether using combinations of various novel stroke rehabilitations can synergistically improve motor recovery [8].

Scalp acupuncture (SA) is a specialized acupuncture technique in which a filiform needle is used to penetrate specific stimulation areas on the scalp [9]. Baihui (GV20)-based SA has been shown to improve infarct volume and neurological function scores and exhibit potential neuroprotective roles in experimental ischemic stroke [10]. SA is commonly used during the acute, recovery, and sequelae stages of ischemic and hemorrhagic strokes [11,12,13,14].

Noninvasive brain stimulation (NIBS) techniques can be used to monitor and modulate the excitability of the intracortical neuronal circuits [15]. Repetitive transcranial magnetic stimulation (rTMS) is a noninvasive method that can change the excitability of the brain cortex for at least several minutes. The nature of the after-effect depends on the frequency, intensity, and pattern of stimulation [16]. Currently, rTMS is being explored as a novel therapy for modulating cortical excitability to improve the motor function in patients with stroke [17]. High-frequency rTMS (HF-rTMS; more than 5 Hz) applied to the ipsilesional hemisphere facilitates cortical excitability [18]; however, low-frequency rTMS (LF-rTMS; 1Hz or less) applied to the contralesional hemisphere decreases cortical excitability [19,20,21,22,23,24]. Di Pino et al. critically reviewed the interhemispheric competition mechanism of synaptic and functional reorganization after stroke and suggested a bimodal balance-recovery model that links interhemispheric balancing and functional recovery to the structural reserve spared by the lesion [15].

SA and electromagnetic convergence stimulation (SAEM-CS) involves the simultaneous application of SA stimulation of Standard International Acupuncture Nomenclature (SIAN)’s MS6 and MS7 at the upper limb regions of the ipsilesional hemisphere and LF-rTMS over the M1 region’s hot spot (motor cortex at the contralesional hemisphere) [25]. Zhao et al. reported that based on routine rehabilitation treatment, SA plus LF-rTMS could promote white matter tract repair better than SA alone, and that the motor function improvement of the hemiplegic upper limb might be closely related to the rehabilitation of the forceps minor [26]. We compared the efficacies of SAEM-CS combined with conventional stroke rehabilitation therapy (CSRT), SA combined with CSRT, LF-rTMS combined with CSRT, and CSRT alone for motor-function recovery (primary aim) and cognitive function, activities of daily living, walking, quality of life, motor-function recovery, and stroke severity (secondary aims) in inpatients with CI to investigate the synergistic effects of SA and LF-rTMS on CI. 

## 2. Materials and Methods

This study followed the standard protocol items of the Recommendations for Interventional Trials (SPIRIT) and CONSORT statement. Detailed methods of this study have been reported previously [25].

### 2.1. Study Design

This study was an outcome-assessor-blinded, single-center, randomized controlled pilot clinical trial with a 1:1:1:1 allocation ratio. Participants (*n* = 60) who fit the inclusion criteria were randomly allocated to the control group (*n* = 15), SA group (*n* = 15), rTMS group (*n* = 15), or SAEM-CS group (*n* = 15). All groups received CSRT twice per day, five times per week, a total of 30 times over the course of a 3 week hospitalization period at Chonnam National University Hospital. In addition, the SA group received SA therapy, the rTMS group received rTMS therapy, and the SAEM-CS group received SAEM-CS therapy once per day. Outcome measures were determined at baseline (week 0), 3 weeks after the first intervention (Week 3), and 4 weeks after completion of the intervention (Week 7). The study design is summarized in Table 1.

### 2.2. Ethical Considerations

This study was conducted in accordance with the Declaration of Helsinki and was approved by the Institutional Review Board of Chonnam National University Hospital (CNUH-2015-114). This trial was registered at cris.nih.go.kr (registration number: KCT0001768). All patients provided written informed consent before participating in this study.

### 2.3. Participant Recruitment

To achieve adequate participant enrolment to reach the target sample size, all CI patients who finished treatment for early acute-stage CI at the Department of Neurology of Chonnam National University Hospital were screened by physical and rehabilitation medicine doctors. Patients who received an explanation regarding this study from the clinical research coordinator (CRC) and who voluntarily signed a consent form were transferred to the Department of Physical and Rehabilitation Medicine to participate in this study. The CRC continuously monitored the medical conditions of enrolled participants for improved adherence to intervention protocols. 

### 2.4. Participation

There were six inclusion criteria: (1) age older than 19 years; (2) incipient CI confirmed by computed tomography or magnetic resonance imaging examination; (3) CI that resulted in motor and sensory disorders within 1 month before enrolment; (4) could undergo rehabilitation therapy after hospitalization at the Department of Physical and Rehabilitation Medicine of Chonnam National University Hospital; (5) modified Rankin scale (mRS) score of 2–4; and (6) voluntarily signed an informed consent form.

Subjects whose general condition was not fit for SA and rTMS therapies were excluded. Detailed exclusion criteria were as follows: (1) prior history of brain lesion (e.g., stroke, serious mental illness, loss of consciousness accompanied by head trauma, brain surgery, or seizure disorder); (2) presence of other serious illnesses (e.g., cancer, Alzheimer’s disease, epilepsy, head trauma, or cerebral palsy); (3) transient ischemic attack; (4) contraindications to electromagnetic stimulation (e.g., metal implants in the brain, implanted electronic devices in the body such as nondetachable ferromagnetic metals, metal-sensitive implants less than 30 cm away from the brain such as cochlear implants, pacemakers, aneurysm clips or coils, stents, bullet fragments, deep brain stimulation, vagus nerve stimulators, jewelry, or hairpins); (5) continuous convulsion symptoms; (6) previous craniectomy or shunt surgery; (7) increased intracranial pressure symptoms such as headache, vomiting, or nausea; (8) seizure disorder or epilepsy after CI; (9) prior history of stroke accompanied by a clear clinical sign; (10) contraindications to SA (e.g., scalp scarring, inflammation from scalp injury, infection in the treatment region, inability to stop blood flow due to clotting disturbances such as hemophilia, serious unusual response after acupuncture treatment); (11) pregnant or breastfeeding; (12) disagreement with informed consent; and (13) scheduled for surgery within 2 weeks.

### 2.5. Randomization and Blinding

After signed informed consent and baseline measurements were obtained, random allocation software (developed by M. Saghaei, MD, Department of Anesthesia, Isfahan University of Medical Sciences, Isfahan, Iran) was used to assign a serial number to the 60 research volunteers and to randomly allocate 15 of them to each group. The serial number codes were inserted in sealed opaque envelopes, kept in a double-locked cabinet, and opened in the presence of the patient and a guardian.

We had no choice but to adopt a single-outcome-assessor blinding approach because sham treatment was impossible due to the nature of SA, which included scalp penetration. During the study, the assessor was blinded to group assignments, and data analysts without conflicts of interest were involved in this study.

### 2.6. Implementation

A CRC was used to generate the allocation sequence, enrol participants, and assign participants to interventions.

### 2.7. Intervention

All participants underwent CSRT, which focused on practicing fine and gross motor movements, activities of daily living, task-oriented therapeutic exercises, and muscular electrical stimulation therapy as needed. Training for swallowing and improving language was also performed for dysarthria. These sessions were conducted for 30 min (excluding Saturdays and Sundays) twice daily for 3 weeks to a total 30 times. SA, rTMS, and SAEM-CS therapies were conducted once daily for 20 min (excluding Saturdays and Sundays) for 3 weeks to a total of 15 times.

SA was conducted as follows: one or two needles were horizontally inserted approximately 3 cm into the lesion site and upper limb regions of MS6 (line connecting GV21 and GB6) and MS7 (line connecting GV20 and GB7) in the directions from GV21 to GB6 and from GV20 to GB7 [14]. Manual stimulation and electroacupuncture were not applied, and the needles (KOS 92 nonmagnetic steel acupuncture needles, size 0.25 mm × 30 mm, product no. A84010.02; Dongbang Acupuncture, Inc., Boryeong, Republic of Korea) were left in position for 20 min.

The rTMS was conducted as follows: a 70 mm figure-8 coil and a Magstim Rapid stimulator (Magstim Co., Dyfed, UK) were used to deliver 1 Hz of rTMS to the skull of the contralesional hemisphere at the site that elicited the largest motor-evoked potentials (MEPs) in the first dorsal interosseous (FDI) muscle of the unaffected upper limb. One LF-rTMS session consisted of 1200 pulses and lasted for 20 min. Stimulation intensity was set to 80% of the motor threshold of the FDI muscle, which was defined as the lowest intensity of stimulation that provoked MEPs. All patients sat in a reclining wheelchair and were asked to relax as much as possible with their heads strapped to a headrest [27].

The SAEM-CS was conducted as follows: the aforementioned SA and LF-rTMS therapies were performed simultaneously. After SA treatment of MS6 and MS7 on the lesion side, LF-rTMS stimulation was conducted on the contralateral hemisphere for 20 min.

### 2.8. Outcome Measurements

The primary outcome was motor function, and the secondary outcomes were cognitive function, activities of daily living, walking, quality of life, recovery of motor function, and stroke severity. Primary and secondary outcome measurements were conducted at baseline (before intervention), 3 weeks after the first intervention, and 4 weeks after completion of intervention (except Korean Mini Mental State Examination (K-MMSE), American Speech–Language–Hearing Association National Outcome Measurement System Swallowing Scale (ASHA-NOMS), and functional ambulatory category (FAC)). The time point of the primary outcome endpoint was 7 weeks after the first intervention.

The primary outcome was assessed via changes in the Fugl–Mayer assessment (FMA) scale scores for motor function. The FMA scale was developed as the first quantitative evaluation instrument for measuring sensorimotor stroke recovery and includes an assessment of the upper extremities (33 items; score range 0–66) and lower extremities (17 items; score range 0–34) [28].

Secondary outcome measures were assessed via changes in the National Institutes of Health Stroke Scale (NIHSS) score, modified Barthel index (MBI), functional independent measurement (FIM) score, K-MMSE score, ASHA-NOMS score, FAC, European Quality of Life-5 Dimensions (EQ-5D), modified Ashworth scale (MAS) score, hand grip strength test, MEPs, mRS score, and 9 hole peg test (9HPT). 

The NIHSS, which was developed by the United States National Institutes of Health, is a standardized stroke severity scale used to describe the neurological deficits of stroke patients, and it strongly predicts the likelihood of a patient’s recovery after stroke [29]. The MBI is a scale that measures 10 basic aspects of daily life activities related to self-care and mobility [30]. The FIM is an assessment of everyday movement performance that evaluates 13 detailed items of motor FIM and 5 detailed items of cognitive FIM [31]. The MMSE is a brief, 30 point questionnaire that is used to screen for cognitive impairment. In this study, we used the K-MMSE [32]. The ASHA-NOMS is a seven stage dysphagia scale developed by the American Speech–Language–Hearing Association to evaluate the severity of dysphagia [33]. The FAC was designed to evaluate walking ability, which is categorized into six ranks [34]. The EQ-5D is a generic instrument for describing and valuing health-related quality of life [35]. The MAS assesses muscles by measuring spasticity in the wrist and elbow joints while the joints are maximally bent [36]. The hand grip strength test evaluates muscle strength in the hands [21].

In this study, MEPs were evoked by stimulating the primary motor cortex representing hand grip muscles without pain. Responses of the FDI muscle were then observed. MEPs are useful for predicting functional recovery after CI. The latency and amplitude of the MEP responses were recorded [37]. The mRS is a six point ordinal hierarchical scale that describes global disability and focuses on mobility [38]. The 9HPT is useful for measuring the dexterity of relatively well-recovered patients [39].

### 2.9. Sample Size Calculation

Because of the lack of adequate preliminary studies and limited research funds, study period, and recruitment opportunities, we adopted a pilot study design with 15 participants in each group. Sample size calculation was detailed in our previously published study protocol [25].

### 2.10. Statistical Analyses

With the approval of the IRB, the statistical analysis was revised from the study protocol. We performed per-protocol analysis (PP group) for the assessment of efficacy; thus, only subjects who completed the three evaluations were analyzed as described in the protocol. All statistical analyses were performed by blinded biostatisticians using SPSS version 20.0 software (SPSS Inc., Chicago, IL, USA) using two-sided significance tests with a 5% significance level. Continuous variables have been presented as means and standard deviations (SD), and categorical variables have been presented as count frequencies and percentages.

Baseline data were collected and compared using the independent k-sample Kruskal–Wallis test and χ^2^ test. Differences between all outcome value changes in the four groups were compared via repeated measures analysis of variance (ANOVA) (Friedman tests). Values of FMA upper extremity (FMAUE), FMA lower extremity (FMALE), FMA total (FMAT), NIHSS, MBI, FIM, 9HPT, mRS, EQ-5D, K-MMSE, MAS elbow, and MAS ankle were compared by repeated-measures ANOVA across two to three testing time points (Week 0, Week 3, Week 7). The Scheffé post hoc test was conducted to detect differences between therapies. Differences between two groups of outcome value changes (Week 0 vs. Week 3 and Week 0 vs. Week 7; significant changes were observed in ANOVA and the Scheffé post hoc test) were compared via the Mann–Whitney U test (nonparametric test).

## 3. Results

### 3.1. Participants

We recruited participants between 31 July 2015, and 31 December 2017. During the study period, 2200 patients were assessed for eligibility and 2140 were excluded due to nonconformity to the inclusion criteria, conformity to the exclusion criteria, or refusal to participate. Sixty patients were included in this study and were randomly assigned to four groups: control group, 15; SA group, 15; rTMS group, 15; and SAEM-CS group, 15. Three did not complete treatment in the control group. Two did not complete treatment and two were lost to follow-up in the SA group. One exited the study due to orthopedic surgery, four did not complete treatment, and two were lost to follow-up in the rTMS group. One exited the study due to orthopedic surgery, two did not complete treatment, and one was lost to follow-up in the SAEM-CS group (Figure 1). Data for 42 CI patients were used in the final analysis.

### 3.2. Baseline Characteristics

Participants were divided into the control group (*n* = 12), SA group (*n* = 11), rTMS group (*n* = 8), and SAEM-CS group (*n* = 11). Baseline demographic characteristics of the 42 CI patients in the four groups, including sex, age, lesion site, and all variables, are presented in Table 2. No significant differences in the baseline demographic characteristics were detected among the four groups (*p* > 0.05; Table 2).

### 3.3. Efficacy of Primary and Secondary Outcomes

#### 3.3.1. Changes in Outcome Measures in the Four Groups

After 3 weeks of intervention, we observed significant improvements in the SAEM-CS group (changes in the FMAUE, FMALE, FMAT, MBI, FIM, 9HPT, and EQ-5D scores), SA group (changes in the FMAUE, FMALE, FMAT, NIHSS, MBI, and FIM scores), rTMS group (changes in the FMAUE, FMALE, FMAT, NIHSS, MBI, FIM, 9HPT, mRS, EQ-5D, and APB recording cortical stim amplitude score), and the control group (changes in the FMAUE, FMALE, FMAT, MBI, FIM, and 9HPT scores; Table 3; all data are provided in Appendix A).

#### 3.3.2. Comparisons of Value Changes in Outcome Measures among the Four Groups

Repeated-measures ANOVA showed a significant interaction between time and group with respect to FMAUE (F = 3.82; *p* = 0.002), FMAT (F = 3.15; *p* = 0.008), MBI (F = 4.27; *p* = 0.001), FIM (F = 3.06; *p* = 0.010), and EQ-5D (F = 4.52; *p* = 0.014; Table 4). Changes in the FMAUE scores (Week 0 vs. Week 3) of the rTMS group were significantly larger than those of the SAEM-CS group, and the changes in the FMAUE scores (Week 0 vs. Week 7) of the rTMS group were significantly larger than those of the control and SAEM-CS groups according to the Scheffé post hoc test (Table 4). Changes in the FMAT scores (Week 0 vs. Week 7) of the rTMS group were significantly larger than those of the control and SAEM-CS groups, according to the Scheffé post hoc test (Table 4).

Changes in the MBI scores (Week 0 vs. Week 3) of the rTMS group were significantly larger than those of the control group, and the changes in the MBI scores (Week 0 vs. Week 7) of the rTMS group were significantly larger than those of the control and SA groups, according to the Scheffé post hoc test (Table 4). Changes in the FIM scores (Week 0 vs. Week 7) of the rTMS group were significantly larger than those of the control and SA groups according to the Scheffé post hoc test (Table 4).

#### 3.3.3. Multiple Comparisons of FMAUE, FMAT, MBI, FIM, and EQ-5D among the Four Groups

We conducted multiple comparisons of FMAUE, FMAT, MBI, FIM, and EQ-5D; significant interactions between time and group were observed in the ANOVA and Scheffé post hoc tests used to investigate the synergistic effects of SA and rTMS.

Changes in the MBI (*p* = 0.005) and FIM (*p* = 0.03) (Week 0 vs. Week 3) scores of the rTMS group were significantly larger than those of control group. Changes in FMAUE (*p* = 0.026) and MBI (*p* = 0.043) (Week 0 vs. Week 3) scores of the rTMS group were significantly larger than those of SAEM-CS group. Changes in FIM (*p* = 0.004) (Week 0 vs. Week 3) scores of rTMS group were larger than those of SA group. Changes in FAMUE (*p* = 0.050) (Week 0 vs. Week 3) scores of SA group were larger than those of SAEM-CS group. (Table 5). 

Changes in FMAUE (*p* = 0.015), FMAT (*p* = 0.023), MBI (*p* = 0.002), and FIM (*p* < 0.001) scores (Week 0 vs. Week 7) of the rTMS group were significantly larger than those of the control group. Changes in FMAUE (*p* = 0.016), FMAT (*p* = 0.012), MBI (*p* = 0.026), and FIM (*p* = 0.012) scores (Week 0 vs. Week 7) of the rTMS group were also significantly larger than those of the SAEM-CS group. Changes in MBI (*p* = 0.016) and FIM (*p* = 0.008) scores (Week 0 vs. Week 7) of the rTMS group were significantly larger than those of the SA group (Table 5).

### 3.4. Safety Evaluation

Adverse events that occurred in this study were recorded on a case report form after evaluating their relationships with the intervention. No adverse events that were related to the intervention occurred in this study. 

## 4. Discussion

To our knowledge, this is the first randomized controlled study to investigate the synergistic effects of SA and rTMS on motor-function recovery, stroke severity, activities of daily living, cognitive function, dysphagia, walking ability, quality of life, and spasticity of CI patients by comparing the effects of simultaneous application of LF-rTMS and SA with the effects of SA, LF-rTMS, and CSRT. There were several main findings. First, rTMS combined with CSRT led to better improvements in FMA, MBI, and FIM than CSRT alone or SAEM-CS combined with CSRT. Second, SA combined with CSRT and SAEM-CS combined with CSRT did not lead to significant differences compared with CSRT alone. Third, SAEM-CS did not show the positive synergistic effects of SA and rTMS on motor-function recovery, stroke severity, activities of daily living, cognitive function, dysphagia, walking ability, quality of life, and spasticity of CI patients.

The rTMS group showed better effects on motor-function recovery and activities of daily living than the control group and SAEM-CS group at 4 weeks after the intervention; however, there were no significant differences in outcome score changes except for MBI and FIM (Week 0 vs. Week 3) between the rTMS group and the control group. These results were similar to those of a previous study [22] and may be related to the long-term effects of rTMS on stroke.

LF-rTMS can inhibit cortical excitability in the stimulated hemisphere, facilitate excitatory interhemispheric balance, increase contralesional hemisphere excitability, and decrease interhemispheric inhibition to promote the recovery of motor function [40]. Several clinical trials have reported no significant effects of LF-rTMS on upper limb motor-function recovery [41,42,43]. However, other studies have confirmed that rTMS produces significant long-term effects that promote the functional reconstruction of the brain neural network and play a lasting regulatory role in modulating cortical excitability at the stimulation site and remote areas [44,45,46]. Long-term effects are more important than short-term effects because long-lasting beneficial effects of rTMS on upper limb motor function are more reliable indicators of successful clinical intervention. Our results showed that LF-rTMS may have long-lasting therapeutic effects on upper limb motor-function recovery and activities of daily living for patients with CI.

There were no significant differences in primary and secondary outcome score changes between the SA group and control group. These results may have been related to acupoint specificity, acupuncture manipulation, and electrical stimulation.

Acupuncture is a complex intervention involving both specific and nonspecific factors associated with therapeutic benefits. Apart from needle insertion, issues such as needling sensation, psychological factors, acupoint specificity, acupuncture manipulation, and needle duration also have relevant influences on the therapeutic effects of acupuncture [47].

The selection and compatibility of acupoints are considered to have a direct effect on the therapeutic effects. According to the concept of “holism” in traditional Chinese medicine, acupoints in limbs, especially those located below the elbow and knee joints, are very important for managing organ and meridian diseases. These points can be therapeutic for local problems and for the whole body [48]. A systematic review of reports of acupuncture treatment for CI revealed 24 studies that used both SA and body acupuncture, 28 studies that used body acupuncture, and 4 studies that used only SA [49]. Both SA and body acupuncture have been used during clinical trials that reported positive effects of acupuncture for ischemic stroke [50,51].

When SA is used to treat stroke patients, manipulation or electroacupuncture (acupuncture combined with electrical stimulation) are usually used to reinforce the therapeutic effects of SA. There are some methods of reinforcing–reducing acupuncture manipulations in traditional Chinese medicine. In clinical practice, mastering the reinforcing–reducing manipulations of acupuncture will contribute to improvements in therapeutic effects [52]. With various factors of manipulation, including lifting–thrusting, twirling–rotating, and variations in the direction, angle, and depth of needle insertion, it is possible to affect the outcomes of acupuncture treatment [53,54]. In most systematic reviews reporting acupuncture for neurogenesis with experimental ischemic stroke [55], Baihui (GV20)-based SA for experimental ischemic stroke [10] and SA for stroke recovery in randomized controlled trials [14], electroacupuncture or manipulation of twirling (needles should be twirled more than 200 times per minute) has been applied. 

This study aimed to investigate the synergistic effects of SA and rTMS on stroke. Therefore, the same acupuncture treatment method used for the SAEM-CS approach had to be used for SA therapy. Subsequently, we could not use the combination of SA and body acupuncture, manipulation, and electroacupuncture to reinforce the therapeutic effects of SA in the SA group. 

There were no significant differences in primary and secondary outcome score changes between the SAEM-CS group and the control group. Changes in FMAUE, FMAT, MBI, and FIM scores of the SAEM-CS group were significantly smaller than those of the rTMS group. These results showed that SAEM-CS may have no positive synergistic effects of SA and rTMS. There could be several reasons that SAEM-CS did not show a positive synergistic effect with SA and rTMS in our study. First, simultaneous bilateral stimulation of SA and rTMS may reduce synergistic effects due to homeostatic metaplasticity. Homeostatic metaplasticity, which stabilizes the activity of neurons and neural circuits, can either augment or reduce synergistic effects, depending on the timing of combination therapy and types of neurorehabilitation that are used [8]. Homeostatic plasticity has been reported to occur when both excitability-changing protocols were applied simultaneously [56]. Second, in our study, we selected patients hospitalized within 1 month after acute stroke to increase the homogeneity of the experiment population. Two studies have reported that noninvasive brain stimulation might have no effect on motor recovery for some acute-phase stroke patients [41,57]. Zhang et al. reported that subjects of most studies investigating the efficacy of LF-rTMS for stroke-induced upper limb motor deficits had chronic subacute stroke [58].

Our study had some limitations. First, our trial was a pilot study with a small sample size, and we lost some subjects for various reasons. Therefore, the number of subjects included in the final analysis was small. Second, we performed per-protocol analysis, not a full analysis set, and our study had a high potential for bias because of the high dropout rate. Our inclusion criterion was inpatient with CI; therefore, participants received treatments for the 3 week hospitalization period at Chnonam National University Hospital. The dropout rate was high because there were many early discharges before the end of intervention. As our study was a pilot study, the data were not sufficient to give information on the efficacy of SA, rTMS, and SAEM-CS on CI. They could, however, indicate whether it is feasible to recruit and randomize participants to a trial of SA, rTMS, and SAEM-CS for stroke. In the future studies, the inclusion criteria will not be limited to inpatients, and the outcome measurements will be simplified to increase adherence to protocol. Third, according to our study design, we did not perform outcome measurements of K-MMSE, ASHA-NOMS, and FAC at Week 7 and did not record the somatosensory evoked potential (SEP) concerning SA stimulation. Therefore, we did not explore the long-term additional effects on cognitive function, dysphagia, and walking and could not exclude any biasing effect of pain. Our study was a pilot study to investigate the synergistic effects of SA and rTMS through primary and secondary outcome measurements in inpatients with CI. Among the outcome measurements, the main outcome measurement was motor-function recovery. It took a lot of time to evaluate the efficacy, because there were many outcome measurements. A long evaluation time was painful for elderly CI patients. Hence, we did not perform outcome measurements of K-MMSE, ASHA-NOMS, and FAC at Week 7 and did not record the SEP concerning SA stimulation. Fourth, we did not investigate synergistic effects of SA and rTMS through various combination methods. We used only simultaneous application of SA at the ipsilesional hemisphere and LF-rTMS over the contralesional hemisphere in combination with SA and rTMS. Further studies of an effective combination of SA and rTMS (i.e., LF rTMS-primed SA or SA-primed LF-rTMS or HF rTMS-primed SA or SA-primed HF-rTMS) should be performed. Fifth, many previous studies exploring the effects of rTMS and SA on motor-function recovery after stroke have focused on subacute and chronic stroke patients [14,58]. However, our study included only patients with acute CI, not patients with subacute and chronic CI.

Many previous studies have suggested that SA and rTMS are effective treatment methods for stroke. We thought that simultaneous application of SA and rTMS might show positive synergistic effects. However, our findings were different from our expectations.

We believe that the results of our study, which investigated the synergistic effects of SA and rTMS on motor-function recovery of patients with stroke, may have varied greatly depending on the subject characteristics (age [59] and level of severity [60]), parameters of rTMS [61] and SA administration [47], and timing of the combination [8]. Therefore, based on our pilot study, large multi-center studies are warranted in the future to confirm the positive synergistic effects of SA and rTMS.

## 5. Conclusions

Several conclusions can be drawn from the results of our study. First, LF-rTMS over the contralesional hemisphere may have long-term therapeutic effects on upper extremity motor-function recovery and on improving activities of daily living. Second, simultaneous application of SA and LF-rTMS did not show the positive synergistic effects of SA and rTMS on motor-function recovery, cognitive function, activities of daily living, walking, quality of life, and stroke severity.

Further studies should investigate the influence of interindividual characteristics on the response to SA and rTMS and the mechanisms of action of each approach. These results are essential for guiding the development of these combined treatment approaches.

## Figures and Tables

**Figure 1 brainsci-10-00087-f001:**
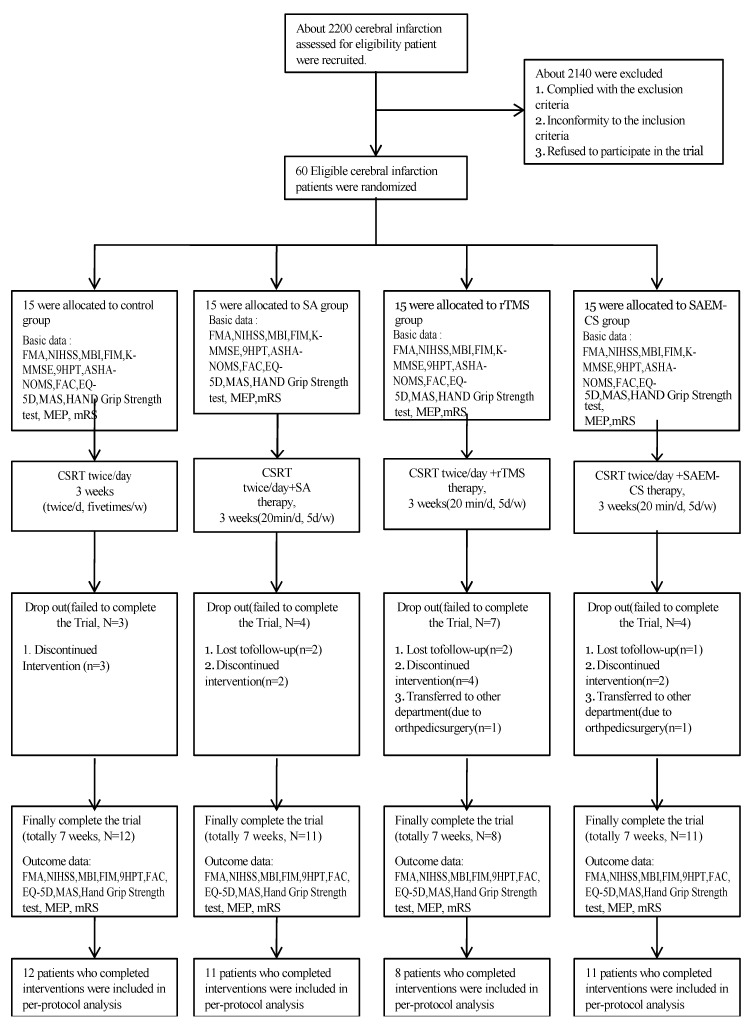
The flow of participants through the trial and the reasons for participant dropout are shown.

**Table 1 brainsci-10-00087-t001:** Enrolment, intervention, and data collection protocols.

Time Point	Enrolment	Allocation	Post-Allocation	Close-Out
Visit_1_	Visit_2_~Visit_6_	Visit_7_~Visit_11_	Visit_12_~Visit_16_	Visit_17_
Week	1	2	3	
Enrolment																		
Informed consent	X																	
Demographic characteristics	X																	
Medical history	X																	
Vital signs	X																	
Inclusion/exclusion criteria	X																	
Random allocation		X	X															
Treatment			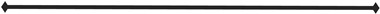
Assessment																		
Change of medical history			X														X	X
Safety assessment			X														X	X
FMA			X														X	X
NIHSS			X														X	X
MBI			X														X	X
FIM			X														X	X
K-MMSE			X														X	
9 HPT			X														X	X
ASHA-NOMS			X														X	
FAC			X														X	
EQ-5D			X														X	X
MAS			X														X	X
Hand grip strength test			X														X	X
MEP			X														X	X
mRS	X		X														X	X

FMA: Fugl-Mayer Assessment; NIHSS: National Institutes of Health Stroke Scale; MBI: Modified Barthel Index; FIM: Functional Independent Measurement; K-MMSE: Korean version of Mini Mental State Examination; 9 HPT: 9-hole peg test; ASHA-NOMS: American Speech-Language-Hearing Association National Outcome Measurement System Swallowing Scale; FAC: Functional Ambulatory Category; EQ-5D: European Quality of Life-5 Dimensions; MAS: Modified Ashworth Scale; MEP: Motor Evoked Potential; mRS: modified Rankin Scale.

**Table 2 brainsci-10-00087-t002:** Homogeneity tests for baseline demographic characteristics and study variables of 42 patients with cerebral infarction.

Dependent Variables	Control Group (*n* = 12)	SA Group (*n* = 11)	rTMS Group (*n* = 8)	SAEM-CS Group (*n* = 11)	*F or* χ^2^ (*p*)
	Mean (SD) or *n* (%)	Mean (SD) or *n* (%)	Mean (SD) or *n* (%)	Mean (SD) or *n* (%)	
Age (y)	62.17 (16.25)	64.45 (14.75)	67.00 (12.92)	67.55 (12.53)	84.69 (0.458) *
Sex (Male)	7 (58.3%)	7 (63.6%)	5 (62.5%)	4 (36.4%)	2.11 (0.550) *
Hemiparesis (Lt side)	8 (66.7%)	5 (45.5%)	5 (62.50%)	7 (63.67%)	1.26 (0.738) *
FMA Upper Extremity	44.42 (26.57)	26.18 (29.20)	33.13 (19.40)	42.64 (27.21)	1.163 (0.337) ^†^
FMA Lower Extremity	19.33 (11.22)	19.27 (8.97)	19.63 (8.77)	19.45 (12.89)	0.01 (1.000) ^†^
FMA Total	63.75 (36.00)	45.45 (36.05)	52.75 (25.02)	62.09 (39.03)	0.66 (0.584) ^†^
NIHSS	3.50 (4.46)	5.73 (3.77)	5.13 (3.23)	4.18 (4.77)	0.62 (0.604) ^†^
MBI	65.58 (18.92)	55.36 (23.08)	41.63 (22.60)	57.55 (30.27)	1.60 (0.205) ^†^
FIM	93.83 (16.72)	87.36 (21.46)	75.38 (13.20)	85.73 (35.99)	0.96 (0.423) ^†^
9HPT	93.65 (39.18)	89.43 (42.73)	108.75 (31.83)	81.90 (42.01)	0.73 (0.538) ^†^
AHSA-NOMS	6.83 (0.58)	6.45 (0.93)	5.38 (1.92)	6.45 (0.82)	3.00 (0.043) ^†^
FAC	2.08 (1.83)	1.18 (1.78)	1.38 (1.51)	1.73 (1.68)	0.60 (0.622) ^†^
mRS	3.08 (0.90)	3.45 (0.69)	3.63 (0.52)	3.27 (0.79)	0.95 (0.425) ^†^
EQ-5D	9.75 (2.73)	9.73 (3.41)	12.13 (1.64)	10.09 (3.27)	1.34 (0.275) ^†^
K-MMSE	26.42 (3.75)	25.00 (4.56)	26.13 (3.52)	23.00 (4.29)	1.56 (0.216) ^†^
MAS elbow	0.08 (0.29)	0.45 (0.93)	0.13 (0.35)	0.27 (0.47)	0.92 (0.442) ^†^
MAS ankle	0.25 (0.62)	0.36 (0.67)	0.00 (0.00)	0.36 (0.50)	0.87 (0.466) ^†^
Grip test, dominant hand	31.17 (18.15)	27.91 (9.68)	25.63 (13.74)	20.55 (9.72)	1.26 (0.302) ^†^
Grip test, non-dominant hand	13.50 (15.56)	10.91 (16.18)	6.50 (9.35)	10.64 (10.27)	0.43 (0.733) ^†^
APB recording cortical stim latency	11.83 (12.43)	8.31 (11.56)	9.45 (13.12)	12.33 (11.86)	0.27 (0.848) ^†^
APB recording cortical stim amplitude	258.33 (412.71)	372.73 (567.00)	188.00 (449.58)	576.73 (781.20)	0.89 (0.458) ^†^
AH recording cortical stim latency	22.68 (22.58)	23.47 (22.63)	20.53 (24.30)	14.52 (20.20)	0.37 (0.778) ^†^
AH recording cortical stim amplitude	294.50 (471.46)	144.09 (181.25)	201.63 (353.65)	238.55 (362.56)	0.35 (0.791) ^†^

The data are presented as the mean (standard deviation) or *n* (percentage %); * The *p* value was obtained by χ2 test; ^†^ The *p* value was obtained via tests for several independent samples: Kruskal–Wallis test. SA: Scalp Acupuncture; rTMS: repetitive Transcranial Magnetic Stimulation; SAEM-CS: Scalp Acupuncture and Electromagnetic Convergence Stimulation.

**Table 3 brainsci-10-00087-t003:** Significant changes in outcome measures after treatment completion in the four groups.

Groups	Dependent Variables	Week 0 (M ± SD)	Week 3 (M ± SD)	Week 7 (M ± SD)	Deference (W3-W0)	Deference (W7-W0)	χ^2^ (*p*)
Control group	FMA upper affected side	44.42 ± 26.57	49.58 ± 24.33	50.25 ± 23.64	5.17 ± 10.53	5.83 ± 11.30	9.15 (0.010)
	FMA lower affected side	19.33 ± 11.22	24.50 ± 10.41	24.67 ± 10.19	5.17 ± 4.17	5.33 ± 6.40	8.83 (0.012)
	FMA total affected side	63.75 ± 36.00	74.08 ± 33.22	74.92 ± 31.13	10.33 ± 12.87	11.17 ± 13.07	11.87(0.003)
	MBI	65.58 ± 18.92	72.42 ± 23.19	81.50 ± 18.53	6.83 ± 13.16	15.92 ± 11.94	17.55 (<0.001)
	FIM	93.83 ± 16.72	101.00 ± 20.53	107.33 ± 17.52	7.17 ± 13.54	13.50 ± 11.44	6.53 (0.038)
	9HPT	93.65 ± 39.18	66.59 ± 35.87	58.72 ± 38.28	−27.06 ± 33.08	−34.93 ± 37.61	12.67 (0.002)
SA group	FMA upper affected side	26.18 ± 29.20	35.27 ± 24.41	39.36 ± 25.24	9.09 ± 8.60	13.18 ± 15.72	14.00 (0.001)
	FMA lower affected side	19.27 ± 8.97	25.27 ± 10.11	24.55 ± 10.53	6.00 ± 4.67	5.27 ± 4.41	9.14(0.010)
	FMA total affected side	45.45 ± 36.05	60.55 ± 32.15	63.91 ± 35.02	15.09 ± 11.89	18.45 ± 17.01	12.05 (0.002)
	NIHSS	5.73 ± 3.77	3.09 ± 3.91	3.36 ± 4.08	−2.64 ± 2.69	−2.36 ± 2.46	9.56 (0.008)
	MBI	55.36 ± 23.08	69.73 ± 29.07	74.00 ± 29.36	14.36 ± 11.74	18.64 ± 18.21	11.46 (0.003)
	FIM	87.36 ± 21.46	98.00 ± 23.23	102.45 ± 23.70	10.64 ± 8.63	15.09 ± 15.12	10.369 (0.006)
rTMS group	FMA upper affected side	33.13 ± 19.40	50.13 ± 10.78	56.50 ± 9.70	17.00 ± 13.89	23.38 ± 14.70	13.61 (0.001)
	FMA lower affected side	19.63 ± 8.77	24.50 ± 5.86	28.25 ± 6.78	4.88 ± 6.49	8.63 ± 5.24	10.13 (0.006)
	FMA total affected side	52.75 ± 25.02	74.63 ± 15.22	84.88 ± 14.24	21.88 ± 17.67	32.13 ± 17.27	12.25 (0.002)
	NIHSS	5.13 ± 3.23	2.88 ± 2.36	2.00 ± 1.85	−2.25 ± 1.75	−3.13 ± 1.73	12.29 (0.002)
	MBI	41.63 ± 22.60	67.38 ± 19.94	85.13 ± 11.68	25.75 ± 10.01	43.50 ± 16.64	15.55 (<0.001)
	FIM	75.38 ± 13.20	97.75 ± 14.74	111.50 ± 8.49	22.38 ± 4.66	36.13 ± 6.88	15.55 (<0.001)
	9HPT	108.75 ± 31.83	83.30 ± 40.88	68.56 ± 37.29	−25.45 ± 34.42	−40.19 ± 35.61	11.27 (0.004)
	mRS	3.63 ± 0.52	2.88 ± 1.36	2.63 ± 1.06	−0.75 ± 1.16	−1.00 ± 0.76	7.52 (0.023)
	EQ-5D	12.13 ± 1.64	10.00 ± 2.45	9.13 ± 1.25	11.00 ± 2.27	10.13 ± 1.46	11.47 (0.003)
	APB recording cortical stim amplitude	188.00 ± 449.58	297.75 ± 459.71	441.75 ± 416.07	109.75 ± 138.34	253.75 ± 573.24	6.35 (0.042)
SAEM-CS group	FMA upper affected side	42.64 ± 27.21	44.91 ± 27.72	47.64 ± 25.32	2.27 ± 7.28	5.00 ± 6.80	7.60 (0.022)
	FMA lower affected side	19.45 ± 12.89	23.64 ± 11.24	25.36 ± 10.57	4.18 ± 5.65	5.91 ± 6.50	10.07 (0.007)
	FMA total affected side	62.09 ± 39.03	68.55 ± 38.39	73.00 ± 35.56	6.45 ± 12.11	10.91 ± 12.43	9.63 (0.008)
	MBI	57.55 ± 30.27	73.09 ± 25.99	80.55 ± 27.19	15.55 ± 14.98	23.00 ± 16.12	17.05 (<0.001)
	FIM	85.73 ± 35.99	101.64 ± 22.55	107.91 ± 23.36	15.91 ± 21.14	22.18 ± 20.84	16.60 (<0.001)
	9HPT	81.90 ± 42.01	63.19 ± 39.08	64.56 ± 45.76	−18.70 ± 36.07	−17.34 ± 51.43	7.00 (0.030)
	EQ-5D	10.09 ± 3.27	9.00 ± 3.13	9.45 ± 3.70	9.73 ± 2.65	10.18 ± 3.19	7.40 (0.025)

**Table 4 brainsci-10-00087-t004:** Results of repeated-measures ANOVA and Scheffé post hoc test for the outcomes of treatment.

Dependent Variables	Source of Variation	SS	df Mean Square		F	*p*	Significant	Scheffé Post hoc Test F (*p*)	
								W3-W0	W7-W0
FMA upper extremity	Time	3043.44	2	1521.72	31.91	<0.001	S	3.68 (0.020) c > d	4.32 (0.010) a < c > d
	Group × Time	1092.10	6	182.02	3.82	0.002	S		
FMA lower extremity	Time	910.07	2	455.03	33.56	<0.001	S	0.23 (0.875)	0.66 (0.580
	Group × Time	64.01	6	10.67	0.79	0.583	NS		
FMA total	Time	7283.95	2	3641.97	47.61	<0.001	S	2.27 (0.096)	4.02 (0.014) a < c > d
	Group × Time	1446.48	6	241.08	3.15	0.008	S		
NIHSS	Time	799.59	2	393.79	40.75	<0.001	S	2.70 (0.059)	2.57 (0.069)
	Group × Time	85.92	6	14.32	1.46	0.203	NS		
MBI	Time	13,331.52	2	6665.76	75.70	<0.001	S	3.51 (0.024) a < c	5.58 (0.003) a < c, b < c
	Group × Time	2254.59	6	375.77	4.27	0.001	S		
FIM	Time	9950.21	2	4975.11	59.62	<0.001	S	2.17 (0.108)	4.36 (0.010) a < c, b < c
	Group × Time	1533.92	6	255.65	3.06	0.010	S		
9HPT	Time	12,277.83	2	6318.91	14.98	<0.001	S	1.63 (0.199)	2.48 (0.076)
	Group × Time	533.35	6	889.73	2.17	0.055	NS		
mRS	Time	10.69	2	5.35	7.71	0.001	S	0.62 (0.607)	0.20 (0.897)
	Group × Time	1.55	6	0.26	0.37	0.894	NS		
EQ-5D	Time	20.51	2	10.25	4.52	0.014	S	0.94 (0.429)	1.00 (0.403)
	Group × Time	36.10	6	6.02	2.65	0.022	S		
MAS elbow	Time	0.96	2	0.48	2.43	0.094	NS	0.95 (0.424)	3.02 (0.041)
	Group × Time	1.65	6	0.28	1.40	0.226	NS		
MAS ankle	Time	0.59	2	0.29	1.71	0.088	NS	1.38 (0.265)	1.11 (0.357)
	Group × Time	1.18	6	0.20	1.15	0.345	NS		
Grip test, dominant hand	Time	52.03	2	26.01	0.85	0.433	NS	1.10 (0.362)	0.60 (0.619)
	Group × Time	152.97	6	25.50	0.83	0.550	NS		
Grip test, non-dominant hand	Time	101.67	2	50.83	1.70	0.189	NS	0.34 (0.795)	0.42 (0.742)
	Group × Time	106.96	6	17.83	0.60	0.732	NS		
APB recording cortical stim latency	Time	234.74	2	117.37	3.07	0.052	NS	1.01 (0.401)	2.37 (0.085)
	Group × Time	319.45	6	53.24	1.39	0.229	NS		
APB recording cortical stim amplitude	Time	842,196.62	2	421,098.31	2.66	0.077	NS	2.08 (0.119)	0.81 (0.495)
	Group × Time	186,621.834	6	301,103.64	1.90	0.092	NS		
AH recording cortical stim latency	Time	130.68	2	66.34	0.40	0.670	NS	0.23 (0.873)	0.68 (0.568)
	Group × Time	679.84	6	113.31	0.70	0.652	NS		
AH recording cortical stim amplitude	Time	26,926.60	2	13,463.30	0.35	0.708	NS	1.10 (0.362)	0.94 (0.430)
	Group × Time	254,560.64	6	42,426.77	1.094	0.374	NS		

a: control group; b: SA group; c: rTMS group; d: SAEM-CS group; df: degrees of freedom; NS: not significant; S: significant.

**Table 5 brainsci-10-00087-t005:** Multiple comparisons of FMAUE, FMAT, MBI, FIM, and EQ-5D scores among the four groups.

Groups	FMAU *p* Value		FMAT *p* Value		MBI *p* Value		FIM *p* Value		EQ-5D *p* Value	
	w3-w0	w7-w0	w3-w0	w7-w0	w3-w0	w7-w0	w3-w0	w7-w0	w3-w0	w7-w0
SA vs. Control	0.419	0.494	0.281	0.459	0.109	0.666	0.497	0.758	0.292	0.171
rTMS vs. Control	0.069	0.015	0.153	0.023	0.005	0.002	0.030	<0.001	0.318	0.666
SAEM-CS vs. Control	0.238	0.757	0.228	0.497	0.277	0.255	0.216	0.267	0.827	0.620
SA vs. SAEM-CS	0.050	0.197	0.087	0.236	0.646	0.669	0.921	0.693	0.286	0.336
rTMS vs. SAEM-CS	0.026	0.016	0.069	0.012	0.043	0.026	0.063	0.012	0.405	0.868
SA vs. rTMS	0.147	0.374	0.321	0.266	0.057	0.016	0.004	0.008	0.966	0.242

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
