# Peer review of "Synergistic Effects of Scalp Acupuncture and Repetitive Transcranial Magnetic Stimulation on Cerebral Infarction: A Randomized Controlled Pilot Trial"

_brainsci, 2020, doi:10.3390/brainsci10020087_

Round 1
Reviewer 1 Report
Introduction needs reordering in its logical consequentiality. In the present form, the rationale of combining SA and rTMS is unclear. I think that it could be better to outline first the concept of neural plasticity as crucial for functional recovery, then illustrate the two main strategies and, last, convince the readers on the rational of such a combined approach. Why did not you carried out correction for multiple comparisons for ANOVA data? Could you please ascertain whether FMA Total Group*Time interaction is significant (table 5)? I think that the ms would benefit from just some more detail on the neurophysiological underpinnings at neural network level of the combined approach that provided patients with the most positive effects. In the limitation section, the authors should add why they did not recorded SEP and/or LEP concerning SA stimulation. In view of that, it is not completely clear to me how the authors can exclude any biasing effect of pain in their protocol and experimental data analysis Minor: lines 40-42 need references. table 1. Pls remove underlined type.Author Response
Response to Reviewer 1 Comments
Point 1: Introduction needs reordering in its logical consequentiality. In the present form, the rationale of combining SA and rTMS is unclear. I think that it could be better to outline first the concept of neural plasticity as crucial for functional recovery, then illustrate the two main strategies and, last, convince the readers on the rational of such a combined approach.
Response 1: Please Thank you for your valuable comments. According to your suggestion, we revised introduction section as follows: “Neural plasticity is the ability of the brain to develop new neuronal connections, acquire new functions, and compensate for impairments. These processes are crucial for motor recovery after stroke [5-7]. Current research aims to determine whether using combinations of various novel stroke rehabilitations can synergistically improve motor recovery [8].
Scalp acupuncture (SA) is a specialized acupuncture technique in which a filiform needle is used to penetrate specific stimulation areas on the scalp [9]. Baihui (GV20)-based SA could improve infarct volume and neurological function scores and exhibit potential neuroprotective roles in experimental ischemic stroke [10]. SA is commonly used during the acute, recovery, and sequelae stages of ischemic and hemorrhagic strokes [11-14].
Noninvasive brain stimulation (NIBS) techniques can be used to monitor and modulate the excitability of intracortical neuronal circuits [15]. Repetitive transcranial magnetic stimulation (rTMS) is a noninvasive method that can change the excitability of the brain cortex for at least several minutes. The nature of the after-effect depends on the frequency, intensity, and pattern of stimulation [16].Currently, rTMS is being explored as a novel therapy for modulating cortical excitability to improve the motor function of stroke patients [17]. High-frequency rTMS (HF-rTMS; more than 5 Hz) applied to the ipsilesional hemisphere reportedly facilitates cortical excitability [18], whereas low-frequency rTMS (LF-rTMS; 1Hz or less) applied to the contralesional hemisphere decreases cortical excitability [19-24]. Di Pino, G et al. critically reviewed the interhemisheric competition mechanism of synaptic and functional reorganization after stroke, and suggested a biomodal balance-recovery model that links interhemispheric balancing and functional recovery to the structural reserve spared by the lesion [15].
SA and electromagnetic convergence stimulation (SAEM-CS) involves the simultaneous application of SA stimulation of Standard International Acupuncture Nomenclature (SIAN)’s MS6 and MS7 at the upper limb regions of the ipsilesional hemisphere and LF-rTMS over the M1 region’s hot spot (motor cortex at the contralesional hemisphere) [25]. Zhao, N et al. reported that based on routine rehabilitation treatment, SA plus LF-rTMS could promote white matter tracts repair better than SA alone and the motor function improvement of the hemiplegic upper limb might be closely related to the rehabilitation of the forceps minor [26]. We compared the efficacies of SAEM-CS combined with conventional stroke rehabilitation therapy (CSRT), SA combined with CSRT, LF-rTMS combined with CSRT, and CSRT alone for motor-function recovery (primary aim) and cognitive function, activities of daily living, walking, quality of life, motor-function recovery, and stroke severity (secondary aims) in inpatients with CI to investigate the synergistic effects of SA and LF-rTMS on CI.” (line 42 – 74)
Point 2: Why did not you carried out correction for multiple comparisons for ANOVA data?
Response 2: Please Thank you for your query. There were concerns that it would be difficult to interpret the study results because there were too many study variables. Therefore, Values of FMAUE, FMA lower extremity (FMALE), FMA total, NIHSS, MBI, FIM, 9HPT, mRS, EQ-5D, K-MMSE, MAS elbow, and MAS ankle were compared by repeated-measures ANOVA across two to three testing time points (week 0, week 3, week 7). The Scheffé post hoc test was conducted to detect differences between therapies. We conducted multiple comparisons of FMAUE, FMAT, MBI, FIM, and EQ-5D significant interaction between time and group were observed in the ANOVA and the Scheffé post hoc test. The results of the Scheffé post hoc test were inserted into table 5. We revised the Statistical analyses section as follows : “Values of FMAUE, FMA lower extremity (FMALE), FMA total, NIHSS, MBI, FIM, 9HPT, mRS, EQ-5D, K-MMSE, MAS elbow, and MAS ankle were compared by repeated-measures ANOVA across two to three testing time points (week 0, week 3, week 7). The Scheffé post hoc test was conducted to detect differences between therapies. Differences between two groups of outcome value changes (week 0 vs. week 3 and week 0 vs. week 7; significant changes were observed in the ANOVA and the Scheffé post hoc test) were compared by the Mann-Whitney U test (nonparametric test).” (line 218-225)
Point 3: Could you please ascertain whether FMA Total Group*Time interaction is significant (table 5)? I think that the ms would benefit from just some more detail on the neurophysiological underpinnings at neural network level of the combined approach that provided patients with the most positive effects.
Response 3: Thank you for your query. The FMA Total Group*Time interaction was significant and changes in FMA Total (week 0 vs. week 7) scores of rTMS group (32.13±17.27) were larger than those of control (11.17±13.07) and SAEM-CS group (10.91±12.43).
Point 4: In the limitation section, the authors should add why they did not recorded SEP and/or LEP concerning SA stimulation. In view of that, it is not completely clear to me how the authors can exclude any biasing effect of pain in their protocol and experimental data analysis
Response 4: Thank you for your comments. Our study was a pilot study to investigate the synergistic effects of SA and rTMS on motor-function recovery (primary aim) and cognitive function, activities of daily living, walking, quality of life, motor-function recovery, and stroke severity (secondary aims) in inpatients with CI. It took a lot of time to evaluate the efficacy, because there were many outcome measurements. A long evaluation time was painful for elderly CI patients. So we did not record Somatosensory Evoked Potential (SEP) concerning SA stimulation. Based on your advice, we revised limitation subsection of the discussion section as follows: “Third, according to our study design, we did not perform outcome measurements of K-MMSE, ASHA-NOMS, and FAC at week 7 and did not record Somatosensory Evoked Potential (SEP) concerning SA stimulation. Therefore, we did not explore the long-term additional effects on cognitive function, dysphagia, and walking and could not exclude any biasing effect of pain. Our study was a pilot study to investigate the synergistic effects of SA and rTMS through primary and secondary outcome measurements in inpatients with CI. Among the outcome measurement, the main outcome measurement is the motor function recovery. It took a lot of time to evaluate the efficacy, because there were many outcome measurements. A long evaluation time was painful for elderly CI patients. So we did not perform outcome measurements of K-MMSE, ASHA-NOMS, and FAC at week 7 and did not record Somatosensory Evoked Potential (SEP) concerning SA stimulation.” (line 377-387)
Point 5: Minor: lines 40-42 need reference. Table 1. Pls remove underlined type
Response 5: Thank you for your comments Based on your advice, we revised table 1, and inserted the reference [3, 4]

Reviewer 2 Report
This is a randomized controlled trial aiming to compare the efficacy of SA, LF-rTMS, and SA and electromagnetic convergence stimulation (SAEM-CS) on motor-function recovery of post-stroke (cerebral infarction) patients as primary objective. It is a valid topic of research, which supports the development of holistic medicine. There are however some major concerns that limit the reporting and potential impact of the findings:
1) This study has a high potential for bias: its endpoint dropout rate is higher than 20% and the dropout difference between groups is higher than 15% (please see https://www.nhlbi.nih.gov/health-topics/study-quality-assessment-tools), which may compromise interpretation and generalizability of outcomes. This limitation and risk for bias must be clearly acknowledged, and authors should suggest strategies for increasing adherence to protocol for future studies.
2) The Introduction section would benefit from further information on the underlying mechanisms – according to current evidence – of both SA and rTMS on stroke; the authors do mention the crucial role of neuroplasticity for motor recovery, but do not clearly establish any evidence-based association with the aforementioned techniques (please see: Di Pino, G. et al. 2014. Nat. Rev. Neurol. doi:10.1038/nrneurol.2014.162; Zhao, N. et al. 2018. J. Integr. Neurosci. DOI: 10.31083/JIN-170043).
3) Apparently, motor-function recovery is also part of the secondary aims, as measured through handgrip strength, MAS, 9HPT, MEPs. The authors should clearly state their study objectives.
4) Why did the authors choose not to apply the K-MMSE, ASHA-NOMS and FAC on the week 7 evaluation?
5) Participants characteristics should include information on stroke site and hemisphere.
6) The Results section is somewhat confusing and needs to be reorganized and presented in accordance with – and in the same sequence as – the aims of the study. I would suggest that the authors divide this section in multiple subsections, and start by presenting outcomes concerning the efficacy of SAEM-CS.
7) The Discussion section needs to be improved and factors that may influence the results and how (SA treatment regimen? rTMS parameters? Population characteristics?) should be considered. Do the authors believe stroke site may impact treatment outcomes of SA, LF-rTMS and/or SAEM-CS?
8) The authors hypothesize that synergistic effects of SA and rTMS may not have been observed because rTMS might not have an effect on motor recovery of some acute stroke phase patients. However, their rTMS only + CSRT group has shown an improvement on the FMA, MBI, and FMI scores, which is somewhat inconsistent with the aforementioned hypothesis.
Minor comments:
9) Line 61: Please give the full name for the acronym SIAN.
10) Lines 101-102: Did the authors mean motor and sensory disorders a month before study onset (study onset = enrollment)?
11) Line 165-166: What did the authors mean by “The time point of the primary outcome endpoint was 3 weeks after the first intervention”? Outcome measures on week 7 were not taken into consideration?
12) Line 209: Please write FMA Upper Extremity in addition to the acronym FMAUE.
13) I would suggest putting the STRICTA checklist on supplementary material. Please add legends to the tables.
14) The Conclusion section should be shorter and really focus on the main evidence and future directions drawn from this study.
Author Response
Response to Reviewer 2 Comments
This is a randomized controlled trial aiming to compare the efficacy of SA, LF-rTMS, and SA and electromagnetic convergence stimulation (SAEM-CS) on motor-function recovery of post-stroke (cerebral infarction) patients as primary objective.
It is a valid topic of research, which supports the development of holistic medicine. There are however some major concerns that limit the reporting and potential impact of the findings:
Point 1: This study has a high potential for bias: its endpoint dropout rate is higher than 20% and the dropout difference between groups is higher than 15% (please see http://www.nhibi.nih.gov/health-topic/study-quality-assessment-tools), which may compromise interpretation and generalizability of outcomes. This limitation and risk for bias must be clearly acknowledged, and authors should suggest strategies for increasing adherence to protocol for future studies.
Response 1: Thank you for your valuable comments. As our study was a pilot study, the data was not sufficient to give information on the efficacy of SA, rTMS, and SAEM-CS on CI. It was able to give an indication of whether it is feasible to recruit and randomize participants to a trial of SA, rTMS, and SAEM-CS for stroke. Based on your advice, we revised the limitation subsection of discussion section as follows: “Second, we performed per-protocol analysis, not full-analysis set, and our study has a high potential for bias because of high dropout rate. Our inclusion criteria was inpatient with CI, therefore, participants received treatments for 3-week hospitalization period at Chnonam National University Hospital. The dropout rate was high because there were many early discharges before the end of intervention. As our study was a pilot study, the data was not sufficient to give information on the efficacy of SA, rTMS, and SAEM-CS on CI. It was able to give an indication of whether it is feasible to recruit and randomize participants to a trial of SA, rTMS, and SAEM-CS for stroke. In the future studies, the inclusion criteria will not be limited to inpatients and the outcome measurements will be simplified to increase adherence to protocol.” (line 369-377)
Point 2: The introduction section would benefit from further information on the underlying mechanisms- according to current evidence-of both SA and rTMS on stroke, the authors do mention the crucial role of neuroplasticity for motor recovery, but do not clearly establish any evidence-based association with the aforementioned techniques (please see :Di Pino, G. et al. 2014. Nat.Rev,Neurol. Doi:10.1038/nrneurol.2014.162: Zhao, N et al.2018. J Integr. Neurosci. DOI: 10.31083/JIN-170043)
Response 2: Thank you for your valuable comments. Based on your advice, we revised the introduction section as follows: “Noninvasive brain stimulation (NIBS) techniques can be used to monitor and modulate the excitability of intracortical neuronal circuits [15]. Repetitive transcranial magnetic stimulation (rTMS) is a noninvasive method that can change the excitability of the brain cortex for at least several minutes. The nature of the after-effect depends on the frequency, intensity, and pattern of stimulation [16].Currently, rTMS is being explored as a novel therapy for modulating cortical excitability to improve the motor function of stroke patients [17]. High-frequency rTMS (HF-rTMS; more than 5 Hz) applied to the ipsilesional hemisphere reportedly facilitates cortical excitability [18], whereas low-frequency rTMS (LF-rTMS; 1Hz or less) applied to the contralesional hemisphere decreases cortical excitability [19-24]. Di Pino, G et al. critically reviewed the interhemisheric competition mechanism of synaptic and functional reorganization after stroke, and suggested a biomodal balance-recovery model that links interhemispheric balancing and functional recovery to the structural reserve spared by the lesion [15].
SA and electromagnetic convergence stimulation (SAEM-CS) involves the simultaneous application of SA stimulation of Standard International Acupuncture Nomenclature (SIAN)’s MS6 and MS7 at the upper limb regions of the ipsilesional hemisphere and LF-rTMS over the M1 region’s hot spot(motor cortex at the contralesional hemisphere) [25]. Zhao, N et al. reported that based on routine rehabilitation treatment, SA plus LF-rTMS could promote white matter tracts repair better than SA alone and the motor function improvement of the hemiplegic upper limb might be closely related to the rehabilitation of the forceps minor [26].”(line 51-69)
Point 3: Apparently, motor-function recovery is also part of the secondary aims, as measured through handgrip strength, MAS, 9HPT, MEPs. The authors should clearly state their study objectives.
Response 3: Thank you for your comments. Based on your advice, we revised the introduction section and outcome measurements subsection of material and methods section as follows: “We compared the efficacies of SAEM-CS combined with conventional stroke rehabilitation therapy (CSRT), SA combined with CSRT, LF-rTMS combined with CSRT, and CSRT alone for motor-function recovery (primary aim) and cognitive function, activities of daily living, walking, quality of life, motor-function recovery, and stroke severity (secondary aims) in inpatients with CI to investigate the synergistic effects of SA and LF-rTMS on CI.” (line 69-74)
“The primary outcome was motor function, and the secondary outcomes were cognitive function, activities of daily living, walking, quality of life, motor-function recovery, and stroke severity.” (line 168-169)
Point 4: Why did the authors choose not to apply the K-MMSE, ASHA-NOMS and FAC on the week 7 evaluation?
Response 4: Thank you for your query. Our study have many outcome measurement, so it took a lot of time to evaluate the efficacy. Among the outcome measurement, the main outcome measurement is the motor function recovery. Therefore, we did not conducted to evaluate cognitive function, dysphagia, and walking ability at follow up (week 7). Based on your advice, we revised limitation subsection of the Discussion section as follows: “Third, according to our study design, we did not perform outcome measurements of K-MMSE, ASHA-NOMS, and FAC at week 7 and did not record Somatosensory Evoked Potential (SEP) concerning SA stimulation. Therefore, we did not explore the long-term additional effects on cognitive function, dysphagia, and walking and could not exclude any biasing effect of pain. Our study was a pilot study to investigate the synergistic effects of SA and rTMS through primary and secondary outcome measurements in inpatients with CI. Among the outcome measurement, the main outcome measurement is the motor function recovery. It took a lot of time to evaluate the efficacy, because there were many outcome measurements. A long evaluation time was painful for elderly CI patients. So we did not perform outcome measurements of K-MMSE, ASHA-NOMS, and FAC at week 7 and did not record Somatosensory Evoked Potential (SEP) concerning SA stimulation.” (line 377-387)
Point 5: Participants characteristics should include information on stroke site and hemisphere.
Response 5: Thank you for your comments. Based on your advice, the left side hemiparesis item was inserted into the table 3
Point 6: The Results section is somewhat confusing and needs to be reorganized and presented in accordance with-and in the same sequence as- the aims of the study. I would suggest that the authors devide this section in multiple subsection, and start by presenting outcomes concerning the efficacy of SAEM-CS.
Response 6: Thank you for your valuable comments. Based on your advice, we devide results section in multiple subsection as follows :
“3.3. Efficacy of primary and secondary outcomes
3.3.1. changes of outcome measures in four group
After 3 weeks of intervention, we observed significant improvements in SAEM-CS group (changes in the FMAUE, FMALE, FMAT, MBI, FIM, 9HPT, and EQ-5D scores), SA group (changes in the FMAUE, FMALE, FMAT,NIHSS, MBI, and FIM scores), rTMS group (changes in the FMAUE, FMALE, FMAT,NIHSS, MBI, FIM, 9HPT, mRS, EQ-5D,and APB recording Cortical stim Amplitude score), and the control group (changes in the FMAUE, FMALE, FMAT, MBI, FIM, and 9HPTscores; Table 4; all data is in Appendix 1).
3.3.2. Comparisons of value changes of outcome measures between four groups
Repeated-measures ANOVA showed a significant interaction between time and group with respect to FMAUE (F=3.82; p=0.002), FMAT (F=3.15; p=0.008), MBI (F=4.27; p=0.001), FIM (F=3.06; p=0.010), and EQ-5D (F=4.52; p=0.014; Table 5).
3.3.3. Multiple comparisons of FMAUE, FMAT, MBI, FIM, and EQ-5D between four groups
We conducted multiple comparisons of FMAUE, FMAT, MBI, FIM, and EQ-5D significant interaction between time and group were observed in the ANOVA and Scheffé post hoc test to investigate the synergistic effects of SA and rTMS.
Changes in the MBI (p=0.005) and FIM (p=0.03) (week 0 vs. week 3) scores of the rTMS group were significantly larger than those of control group. Changes in FMAUE (p=0.026) and MBI (p=0.043) (week 0 vs. week 3) scores of the rTMS group were significantly larger than those of SAEM-CS group. Changes in FIM (p=0.004) (week 0 vs. week 3) scores of rTMS group were larger than those of SA group. Changes in FAMUE (p=0.050) (week 0 vs. week 3) scores of SA group were larger than those of SAEM-CS group. (Table 6).
Changes in FMAUE (p=0.015), FMAT (p=0.023), MBI (p=0.002), and FIM (p<0.001) scores (week 0 vs. week 7) of the rTMS group were significantly larger than those of the control group. Changes in FMAUE (p=.016), FMAT (p=0.012), MBI (p=0.026), and FIM (p=0.012) scores (week 0 vs. week 7) of the rTMS group were also significantly larger than those of the SAEM-CS group. Changes in MBI (p=0.016) and FIM (p=0.008) scores (week 0 vs. week 7) of the rTMS group were significantly larger than those of the SA group (Table 6).
3.4. Safety evaluation
Adverse events that occurred in this study were recorded on a case report form after evaluating their relationships with the intervention. No adverse events that were related to the intervention occurred in this study.” (line 254-293)
Point 7: The Discussion section needs to be improved and factors that may influence the results and how (SA treatment regimen? rTMS parameters? Population characteristics?) should be considered. Do the authors believe stroke site may impact treatment outcomes of SA, LF-rTMS and/or SAEM-CS?
Response 7: Thank you for your valuable comments. We think SA treatment regimen, rTMS parameter, and polpulation characteristics may influence the efficacy of SA, rTMS, and SAEM-CS on CI. However, It is unclear whether stroke site affects the therapeutic effects. Based on your advice we inserted the following paragraph into the discussion section :
“ We think the results of our study investigating the synergistic effects of SA and rTMS on motor-function recovery of patients with stroke may vary greatly depending on the subject characteristics (age [59] and level of severity [60]), parameters of rTMS [61] and SA administration [47], and timing of the combination [8]). Therefore, based on our pilot studies, large multi-center studies should be conducted in the future to conclude about positive synergistic effects of SA and rTMS.” (line 398-403)
Point 8: The authors hypothesize that synergistic effects of SA and rTMS may not have been observed because rTMS might not have an effect on motor recovery of some acute stroke phase patients. However, their rTMS only + CSRT group has shown an improvement on the FMA, MBI, and FMI scores, which is somewhat inconsistent with the aforementioned hypothesis.
Response 8: Thank you for your comments. We expected that SAEM-CS would have positive synergistic effects of SA and rTMS on acute CI. However, our findings were different from what we expected. We inserted the following paragraph into the discussion section: “Many previous studies have suggested that SA and rTMS were effective treatment methods for stroke. We thought that simultaneous application of SA and rTMS might show positive synergistic effects. However, our findings were different from what we expected.” (line 395-397)
Minor comments
Point 9: Please give the full name for the acronym SIAN
Response 9: Thank you for your comments. We inserted the full name for the acronym SIAN as follows: “SA stimulation of Standard International Acupuncture Nomenclature (SIAN)’s MS6 and MS7” (line 64)
Point 10: Lines 101-102: Did the authors mean motor and sensory disorders a month before study onset (study onset = enrollment)?
Response 10: Thank you for your query. Based on your advice, we revised the inclusion criteria subsection as follows: “(3) CI that resulted in motor and sensory disorders within 1 month before enrollment;” (line 107-108)
Point 11: Line 165-166: What did the authors mean by “The time point of the primary outcome endpoint was 3 weeks after the first intervention” Outcome measures on week 7 were not taken into consideration?
Response 11: Thank you for your comments. This is our mistake. We revised “3 weeks after the first intervention” to “7 weeks after the first intervention” (line 174-175)
Point 12: Line 209: Please write FMA Upper Extremity in addition to the acronym FMAUE.
Response 12 : Thank you for your comments. Based on your advice, We revised it as follows: “Values of FMA upper extremity (FMAUE), FMA lower extremity (FMALE), FMA total (FMAT), ….” (line 218-219)
Point 13: I would suggest putting the STRICTA checklist on supplementary material. Please add legends to the tables.
Response 13: Thank you for your suggestion. Based on your suggestion, we put the STRICTA checklist on supplementary material as a Appendix 2. And added legends to the tables
Point 14: The Conclusion section should be shorter and really focus on the main evidence and future directions drawn from this study.
Response 14: Thank you for your valuable comments. Based on your advice, We revised the conclusion section as follows: “Several conclusions can be drawn from the results of our study. First, LF-rTMS over the contralesional hemisphere may have long-term therapeutic effects on upper extremity motor-function recovery and on improving activities of daily living. Second, simultaneous application of SA and LF-rTMS did not show the positive synergistic effects of SA and rTMS on motor-function recovery, cognitive function, activities of daily living, walking, quality of life, and stroke severity. Further studies should investigate the influence of interindividual characteristics on the response to SA and rTMS and the mechanisms of action of each approach. These results are essential for guiding the development of these combined treatment approaches.” (line 406-414)

Round 2
Reviewer 2 Report
No further comments.
Author Response
Thank you for your comments.
To address your suggestion, our revised manuscript was professionally edited to ensure that the writing is clear and the spelling and grammar are free of errors. The edit was performed by editors at Editage, a division of Cactus Communications. Editage offers professional English language editing and publication support services to authors engaged in over 500 areas of research.